# Using Deep Siamese Neural Networks to Speed up Natural Products Research

## Abstract

Natural products (NPs, compounds derived from plants and animals) are an important source of novel disease treatments. A bottleneck in the search for new NPs is structure determination. One method is to use 2D Nuclear Magnetic Resonances (NMR) spectroscopy, which indicates bonds between nuclei in the compound, and hence is the "fingerprint" of the compound. Computing a similarity score between 2D NMR spectra for a novel compound and a compound whose structure is known helps determine the structure of the novel compound. Standard approaches to this problem do not scale to larger databases of compounds. Here we use deep convolutional Siamese networks to map NMR spectra to a cluster space, where similarity is given by the distance in the space. This approach results in an AUC score that is more than four times better than an approach using LDA.

## 1 Introduction

Natural products (NPs) obtained from both terrestrial and marine organisms are the single most important source of drug leads and new therapeutics. Approximately 70% of all drugs in the clinic today have an origin or inspiration from natural products of plants, animals and microorganisms (Gerwick & Moore, 2012; Mayer et al., 2010; Molinski et al., 2009; Newman & Cragg, 2016). In this regard, NPs have also been a major inspiration for the development of many of the pharmaceutical drugs currently available. This trend is continuing in that NPs, NP derivatives and their mimics, account for roughly 50% of all new drugs over the past several years (Newman & Cragg, 2016). Thus, NPs continue to be an important source of new pharmaceuticals and pharmaceutical leads (Pye et al., 2017).

A major bottleneck in drug discovery is determining the molecular structure of a novel compound. This process is quite time consuming, and although a skilled and experienced NP researcher can be quite effective in this pursuit, most structure elucidations are limited by the poor quality of data, subjective human evaluation of these data, and a step-wise deduction of the structure indicated by this information. Therefore, the structure elucidation of novel chemical entities is oftentimes a long and laborious process. To effectively mine the rich repertoire of chemical diversity of NPs, as well as to identify the diverse NPs present in various other classes of organisms, new automated methods (Leao et al., 2017; Wang et al., 2016; Zhang et al., 2017b) of structural analysis are greatly needed. In fact, some cheminformatics tools developed to meet this need have achieved wide application (Taboada et al., 2017; Tao et al., 2018; Zhang et al., 2017a).

A major technique used for this process is 2D Nuclear Magnetic Resonance (NMR) spectroscopy, and in particular, Heteronuclear single-quantum correlation spectroscopy (HSQC NMR) has been found to be work well for small molecules. This method detects correlations between nuclei of two different types that are linked by one bond. This gives one peak per pair of coupled nuclei, whose two coordinates are the chemical shifts of the two coupled atoms. Three examples are shown in Figure 1. The first two are from the same compound family, and so their spectra are similar. This approach requires very careful analysis that is time consuming, and requires a high level of investigator skill and experience. Our goal is to use machine learning to help accelerate this process. In short, the idea is to learn to map novel NMR spectra into a similarity space where compounds with known structures similar to the novel compound are nearby in the space, thus giving cues to its structure. This technique can also tell us whether the candidate compound is actually one we have seen before, a process that is awkwardly named "dereplication" in this field.

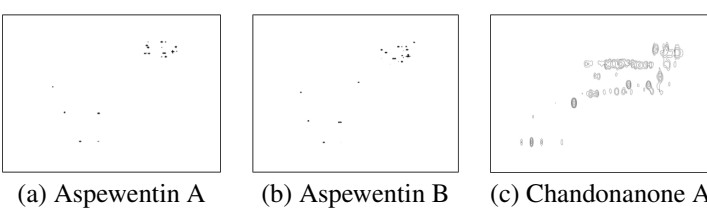

(a) Aspewentin A      (b) Aspewentin B      (c) Chandonanone A

Figure 1: Examples of 2D NMR spectra, molecular "fingerprints."

In this paper, we apply deep siamese convolutional neural networks to this problem. For comparison purposes, we also apply previously used techniques, Probabilistic Latent Semantic Indexing (PLSI) and Latent Dirichlet Allocation (LDA) (Hofmann, 2001; Blei et al., 2003).

## 2 RELATED WORK

A number of techniques are currently being employed for the purpose of performing similarity search of one-dimensional NMR spectra (Barros & Rutledge, 2005; Krishnan et al., 2004; Steinbeck et al., 2003). However, there are very few similarity search techniques for two-dimensional NMR spectra. The most significant technique was proposed by Wolfram et al. (2006). Their approach models two dimensional NMR spectra as documents, and uses topic modeling methods to find similarity between NMR spectra. A redundant grid-based mapping is used to map the peaks in the spectrum to "words." Probabilistic Latent Semantic Indexing (PLSI) (Hofmann, 1999) is used to find the hidden topics in the documents. A cosine similarity function (which operates on the "words" representing the peaks of the spectra) is used to calculate similarity between documents.

The major innovation of Wolfram et al. was their method for mapping of the continuous peaks in spectra such as those in Figure 1 into discrete "words" so that text retrieval techniques can be used. They start with a simple grid-based mapping that overlays the spectrum with a grid, and then uses the grid coordinates of every peak as a word. A major problem with this approach occurs when two nearby peaks end up in different grid cells. If another compound has a similar pair of peaks, but they end up in the same grid cell, the model will miss the similarity. Also, a grid cell may appear in the test set that is not seen in the "vocabulary" of the training set.

To overcome these problems, a redundant grid mapping technique is used that applies multiple, overlapping grid cells with strides that are half of the grid cell width, and multiple grid cell sizes. Now a peak in the spectrum is mapped to the set of grid cells that it is a member of, where each grid cell is identified by four integers: its x,y coordinate as well as the shift amount and the direction from the origin. This becomes the vocabulary of the "document," and PLSI is used to treat the collection of spectra as documents. The similarity of the grid cell approach to convolutional networks is obvious, which motivates our approach.

Wolfram et al. use a real world dataset consisting NMR spectra of ten compound families with approximately 120 compounds. They use leave-one-out cross validation and generate precision-recall curves that are averages from cross validation over all the groups. The most significant part of this paper is the method used to map NMR spectra to documents. This acts as a general framework on which any topic modeling technique can be used to find similarity of NMR spectra. We later apply another topic modeling technique, Latent Dirichlet Allocation (LDA), on the same mapping.

## 3 METHODS

In this section, we describe the data acquisition, more details on the competing approaches we evaluated, and then describe our model, and evaluation metrics.

### 3.1 DATA ACQUISITION AND PROCESSING

We collected 4,105 HSQC NMR spectra from *Journal of Natural Products*, years 2011 through 2015, and *Organic Letters*. Many compound families had fewer than 10 examples, in particular. Examples corresponding to these compound families were removed, and the remaining dataset contained 1,385

NMR spectra from 104 families. We refer to this subset of the data as NAT10, as each compound family has 10 or more compounds.

The spectra were cut-and-pasted from the PDF of the figures in the articles. Each image was saved as a grayscale 512 by 512 pixel PNG image. The images were then manually processed to remove axis ticks, annotations, and unwanted structural formulas that were occasionally included. There was no consistent orientation of the axes across the dataset, so many images were flipped or rotated so as to impose a globally consistent orientation. Each image was then converted to a binary image where 1 represents signal or noise, and 0 represents the background. Each image was labeled by the publication in which it was found, which corresponds to a unique compound family, because we only used data from publications which reported a single new compound family. Finally, some amount of unwanted noise was removed by applying a median cross filter to the images.

The distribution of compounds in the dataset is heavily skewed toward having fewer compounds per class. We found that using compound families with fewer than 10 representatives resulted in too much noise, so all of our experiments were done using the dataset described above.

## 3.2 COMPARISON METHODS

To see how well our Siamese CNN performed, we used two topic modeling techniques: PLSI, as used by Wolfram et al., and Latent Dirichlet Allocation (LDA) (Blei et al., 2003) to measure baseline performance on our dataset. We first mapped our NMR spectra to documents and words as described in Wolfram et al. (2006) to use the topic modeling techniques. For the sake of completeness we give a brief description of PLSI and LDA.

### 3.2.1 PLSI

Probabilistic Latent Semantic Indexing is based on a generative probabilistic model (Hofmann, 1999). The documents are assumed to generate a distribution of aspects (topics) and the topics are assumed to generate a distribution of words. Basically, it introduces a latent topic variable $z$ to explain the relation between words and documents. In effect, one can derive a low-dimensional representation of the observed variables in terms of their affinity to certain hidden variables (topics). After training, each document will have a discrete distribution over all topics, and each topic will have a discrete distribution over all words . PLSI uses a generalization of the Expectation Maximization algorithm to maximize the log-likelihood of the data.

### 3.2.2 LDA

Latent Dirichlet Allocation (Blei et al., 2003) is also an unsupervised generative model that assigns topic distributions to documents. The main difference between PLSI and LDA is in PLSI we smooth $P(w)$ but in LDA we smooth $P(w|z)$ and $P(d|w)$ (where $w$ represents words, $d$ represents documents, and $z$ represents the hidden topic variable). That is, LDA "smooths" PLSI by placing priors on 1) the distribution of topics for a document 2) the distribution of words for a topic. LDA has proven to be better at topic modeling than PLSI (Blei, 2012). We use the same redundant grid-based mapping technique described above and then use LDA to find the hidden topics in the documents. We then use Jensen-Shannon divergence to find the similarity between documents. Jensen-Shannon divergence is based on Kullback-Leibler but is symmetric.

## 3.3 ARCHITECTURE

For this task we use a deep siamese convolutional neural network to learn a metric space of raw NMR spectra interpreted as images (Bromley et al., 1993; Chopra et al., 2005; Hadsell et al., 2006). When all of the training data is projected into this metric space, the result is a searchable cluster map of NMR spectra.

Siamese neural networks, first introduced by Bromley et al. (1993), have been shown to be successful in situations where the number of classes is not known at the time of training, when the number of classes is very large, and when the number of examples per class is very small. All of these properties make them ideal for our application. These models require two identical networks with shared weights (hence the name) trained on pairs of examples where the target is defined by whether

or not the two examples belong to the same class. Due to the nature of training on pairs of examples, the effective size of the training set for a siamese network becomes $\mathcal{O}(F^2)$, where $F$ is the average category size. Consequently, this method is appropriate for situations in which the data is sparse, as is the case for this dataset of NMR spectra.

In order to learn the similarity metric between pairs of images $X_1$ and $X_2$, siamese networks rely on contrastive loss introduced by Hadsell et al. (2006). This is formally defined as

$$L(W, Y, X_1, X_2) = (1 - Y)\frac{1}{2}(D_W(X_1, X_2))^2 + (Y)\frac{1}{2}\{max(0, m - D_W(X_1, X_2))\}^2 \tag{1}$$

where $m$ is a margin, $D_W$ is the Euclidean distance between the outputs of the two networks, parameterized by $W$, and $Y$ is an indicator variable marking whether $X_1$ and $X_2$ are from the same category or not. When $Y = 0$, $X_1, X_2$ are of the same class, and when $Y = 1$, they are from different categories. For this work, we use the compound family to determine "same" and "different." Thus, if the inputs are from the same category, the first term is selected, and the outputs are moved closer together. When they are different, the outputs are moved farther apart, up to the margin. Using $G_W()$ to indicate the output of the network, $D_W$ is simply defined as:

$$D_W(X_1, X_2) = ||G_W(X_1) - G_W(X_2)||_2 \tag{2}$$

Thus the network clusters the inputs in the output space - there is no specific "target" that the network is trying to achieve. The dimensionality of the output of the network determines the dimensionality of the cluster space. For example, the Aspewentins (as in Figure 1) form a single family, so the network tries to map these NMR spectra to nearby points in the output space, and tries to map other compounds farther away.

A slight variation of contrastive loss is used for our system, where a "pushing factor" $P$ is introduced, specifically

$$L(W, Y, X_1, X_2) = (1 - Y)\frac{1}{2}(D_W(X_1, X_2))^2 + (Y)\frac{P}{2}\{max(0, m - D_W(X_1, X_2))\}^2, \tag{3}$$

where $P = 1.5$ is used to more heavily weight the loss for negative examples, which leads to pairs of examples in different classes to be separated more aggressively.

Again, siamese networks have several advantages over a simple classifier network:

- The dataset is "amplified" by using pairs of inputs rather than single ones. Given $n$ members of a category, there are $n(n - 1)/2$ positive examples, and a very large number of negative ones - but these must be balanced, so negative examples are chosen randomly.
- The number of categories does not need to be known in advance; we can use any number of output units for the cluster space;
- Novel examples can be mapped into the cluster space, and in our case, their nearest neighbors should have similar structures.

Our architecture consists of 4 convolutional layers followed by 4 fully connected layers, which is similar to AlexNet. The final layer of dimension $K$ is the dimension of the similarity metric space. This architecture is replicated with tied weights so as to be used in the framework of siamese networks ($G_W(\cdot)$ in the above equation). Specific details regarding the architecture can be found in Table 1.

### 3.4 Training and evaluation

We use a random train, validation, and test split of 80%, 10%, 10%, respectively in which we train using mini-batch stochastic gradient descent along with the Adam optimizer, in which every parameter has its own adaptive learning rate, initialized to 0.001. Mini-batches of size 256 were carefully balanced such that 50% of the mini-batch consists of positive pairs of examples. The tanh activation function is used at all layers except for the output layer, which was linear. Weights were initialized using Xavier initialization. Batch normalization was used after each convolutional layer, as it was found to speed up convergence by a factor of 7. The duration of training was chosen based

Table 1: Architecture

| LAYER | TYPE | FILTERS | DIMENSIONS | ADDITIONAL INFORMATION |
|---|---|---|---|---|
| 1 | Convolutional | 8 (Stride 1) | 4x4 | 2x2 Maxpool |
| 2 | Convolutional | 16 (Stride 1) | 7x7 | 4x4 Maxpool |
| 3 | Convolutional | 16 (Stride 1) | 4x4 | 4x4 Maxpool |
| 4 | Convolutional | 16 (Stride 1) | 4x4 | 4x4 Maxpool |
| 5 | Fully Connected | - | 128 | Dropout=0.5 |
| 6 | Fully Connected | - | 128 | Dropout=0.5 |
| 7 | Fully Connected | - | 128 | Dropout=0.5 |
| 8 | Fully Connected | - | K | - |

$$Precision^{\mu} = \frac{\sum_{i=1}^{|C|} TP_i}{\sum_{i=1}^{|C|}(TP_i+FP_i)} \qquad Recall^{\mu} = \frac{\sum_{i=1}^{|C|} TP_i}{\sum_{i=1}^{|C|}(TP_i+FN_i)}$$

on the performance on the validation set, which was computed every 100 iterations. When loss on the validation set increased 3 times in a row, training was halted, typically resulting in around 10,000 iterations of training. Unless otherwise specified, we used an output dimension of $K = 10$, based on pilot experiments.

We use precision-recall curves measured on examples projected into the learned metric space to evaluate the quality of the embedding. We compute precision and recall by first calculating the mean of each cluster in the training set (where a cluster is defined as members of the same compound family), and computing the euclidean distance between each point in the test set and the cluster means from the training set. Intuitively this is equivalent to varying the threshold of a hypersphere centered at a cluster center while computing precision scores and recall scores of the elements in the test set. In this way, precision can be thought of as the fraction of test points within the hypersphere that match the class of the cluster mean. Similarly, recall can be thought of as the fraction of points matching the class of the cluster mean which are contained within the hypersphere. The results are then micro-averaged over the classes in the training set, meaning that for each threshold, the true positives ($TP$), false positives ($FP$), and false negatives ($FN$) are summed across all of the classes before using the final counts to compute a single precision-recall curve. Micro-averaging for precision and recall is formally defined as

This is distinct from macro-averaging, in which precision and recall are computed for each class, and the resulting precision and recall scores are averaged over the classes. Macro-averaging weights each class the same, so it is less appropriate than the unweighted aggregation used in micro-averaging in situations in which there may be class imbalance, as is the case for the NAT10 dataset.

In order to numerically compare precision-recall curves, we compute the Area Under the precision-recall Curve (AUC) score, which provides us with a single number by which we can compare the quality of a cluster map. A perfect AUC score is 1, as precision and recall are both bounded by 1, and a low AUC score would be close to 0.

We also use the purity measure to evaluate the quality of a cluster map, and as a sanity check on out results (Ezenkwu et al., 2015). The purity measure provides the average number of examples that compose the majority class over each of the clusters. In order to compute this we must first know the cluster assignments for each example. For this reason purity is considered to be an external evaluation criterion, as it requires a notion of an external oracle to define the clusters. If the cluster assignment perfectly matches the class labels for each point, purity will evaluate to 1. In the worst case, purity will be close to 0. The definition of the purity measure is

$$purity(\Omega, \mathbb{C}) = \frac{1}{N} \sum_k \max_j |\omega_k \cap c_j| \tag{4}$$

where $\Omega = \{\omega_1, \omega_2, ..., \omega_{|\Omega|}\}$ represents the set of clusters and $\mathbb{C} = \{c_1, c_2, ..., c_{|\mathbb{C}|}\}$ represents the set of classes.

Due to our knowledge of the ground truth class assignments for each example, we compute the class means for each class, and as if we were computing K-means we assign each point in the metric space to the closest class mean. This gives us a close approximation to what an oracle would provide as the cluster means, and so we use these assignments as $\Omega$, the cluster assignments.

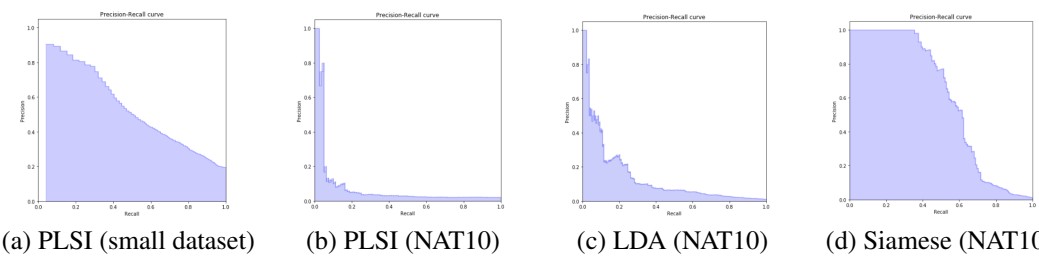

| (a) PLSI (small dataset) | (b) PLSI (NAT10) | (c) LDA (NAT10) | (d) Siamese (NAT10) |

Figure 2: Precision-recall curves for PLSI, LDA, and the Siamese network. Panel (a) is a demonstration that our implementation of PLSI obtains results similar to those in their paper with a similar-sized dataset.

## 3.5 EXPERIMENTS

We performed two experiments. In the first experiment, we separated the training, holdout, and test sets by randomly splitting the 1,385 NMR spectra into 80% training, 10% validation, and 10% test sets. In this case, the compounds in the holdout and test sets will generally have compounds in the same family in the training set. We use the loss on the validation set to stop training, and then report the precision recall curves and the AUC for the test set. We evaluate all three approaches the same way.

The second experiment addresses the actual use-case of the model, where a completely novel set of compounds is discovered. To model this case, we held out four families from training for use in testing: the aphanamixoids, teuvissides, tasiamides, and macrolactins. We call these the "probe families." Evaluation here is more difficult, as there is no ground truth for the result.

Instead, in this case we calculate the averaged Tanimoto score (a measure of structure similarity, closely related to the Jaccard Similarity coefficient, here scaled between 0 and 100) for the top five closest compound families of the probe families, using the PubChem Score Matrix Service (Cai et al., 2014; Lv et al., 2014; Mevers et al., 2014; Mondol et al., 2011; Kim et al., 2016). Ideally, the PubChem website assigns each chemical compound a unique PubChem Compound Identification (CID) number in order to run the Tanimoto score calculation. Unfortunately, not all CIDs of the compounds are retrievable. Furthermore, we observed compounds with incorrect structures in their CID database (e.g., munronins) (Yan et al., 2015). Hence, we only performed the Tanimoto score calculation for aphanamixoids and teuvissides, comparing them to some of their top hits provided by our system whose CIDs were accessible on PubChem. These results are shown in the Supplementary material.

## 4 RESULTS

### 4.1 PERFORMANCE ON HELD OUT DATA

In order to verify our implementation of the grid-based vocabulary and PLSI, we aimed to replicate the results of Wolfram et al. using the same amount of data and the same cross-validation technique as used in their work (Wolfram et al., 2006). Figure 2(a) shows the precision-recall curve on a subset of our dataset with 10 families of compounds and around 120 compounds in the dataset with leave-one-out cross validation. This curve is comparable to the ones in their paper, providing experimental evidence that our implementation of PLSI and redundant grid mapping is correct.

However, using the full NAT10 dataset, we find that PLSI performs quite poorly compared to scenarios in which there are fewer classes and significantly less data. Figure 2(b) shows the precision-recall curve for PLSI trained on NAT10. PLSI performs very well on a small number of families with roughly balanced membership, but fails to generalize to a dataset with a large number of families of various sizes. This indicates that the PLSI technique does not scale to large datasets.

Figure 2(c) shows that LDA outperforms PLSI on the NAT10 dataset. LDA has higher precision at most recall values, whereas PLSI only has high precision when recall is low. We can see from Table 2

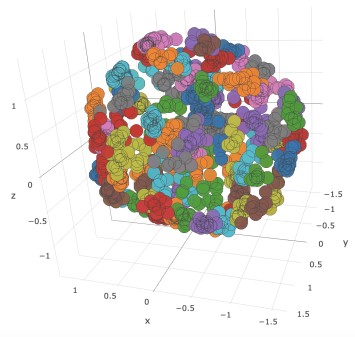

Figure 3: Many NMR spectra projected into a learned metric space of K=3. Matching colors indicate the same compound family, but some of the colors have been repeated for different families.

Table 2: Comparison of AUC scores

| METHOD | AUC SCORE |
|---|---|
| PLSI (10 classes) | 0.51 |
| PLSI | 0.09 |
| LDA | 0.14 |
| Siamese neural network | 0.60 |

that LDA has an AUC score of $0.14$, which is a slight improvement over the AUC score achieved by PLSI.

Figure 2(d) contains the precision-recall curve for our siamese architecture. There is a significant difference in the performance displayed by these curves in favor of our method. Concretely, Table 2 shows that the AUC score for PLSI is $0.09$, while the AUC score for our model is $0.60$. The advantage of the learned representation using the Siamese loss is obvious.

We found that our architecture consistently reached an average final training purity score of $purity(\Omega, \mathbb{C}) = 0.99$, and a validation purity score of $purity(\Omega, \mathbb{C}) = 0.73$ using $K = 10$ for the dimension of the resulting metric space. Figure 3 shows a subset of the NAT10 dataset projected into a more human-friendly 3D metric space learned by the siamese neural network, which clearly shows well defined clusters of the various compound families.

### 4.2 PERFORMANCE ON NOVEL COMPOUND FAMILIES

As mentioned above, one particular advantage of siamese neural networks is the ability to generalize to new classes that it has not been trained on, and in fact, the number of classes that it is expected to support does not have to be known at training time. This property is of considerable interest to us because the real-world domain is such that the number of all classes, compound families in this case, is unknown.

Table 3 shows the top five hits for two compound families, and the similarity (inverse distance in the cluster space) between them and the held out family. Tanimoto scores of 2D chemical structures were calculated in comparison with the similarity score of the NMR spectra of those compounds generated by the siamese neural network model (see Table 3 for the top 5 hits list and Supplementary information for the chemical structures of some of those compounds).

A higher Tanimoto score indicates structure similarity of two compounds. The average intra-cluster Tanimoto score of the cluster containing aphanamixoids C, D, E, F and G is 95.7, and the cluster containing turrapubins A, B, C, D, E, F, G, H, I and J is 84.5. The average intra-cluster Tanimoto score of the cluster containing khayseneganins D, H and 3-deacetylkhivorin is 90.9 (Yuan et al., 2013). All of these intra-cluster Tanimoto scores are higher than the inter-cluster Tanimoto score $T_{(aphanamixoids-turrapubins)} = 70$ or $T_{(aphanamixoids-khayseneganins)} = 74$. However, in contrast to the Tanimoto scores, turrapubins are ranked closer to aphanamixoids than khayseneganins by our system, which is consistent with the fact that $\beta$-furan linked lactone is present in khayseneganins,

Table 3: Top 5 hits for 2 held out families.

| HELD OUT FAMILY | APHANAMIXOIDS | SIMILARITY | TEUVISSIDES | SIMILARITY |
|---|---|---|---|---|
| Rank 1 | turrapubins | 3.56 | oleraceins | 1.30 |
| Rank 2 | munronins | 1.52 | sophodibenzoside s | 1.28 |
| Rank 3 | khayseneganins | 1.42 | aquaterins | 1.15 |
| Rank 4 | Inositol Derivatives | 1.23 | bruceollines and yadanziolides | 1.09 |
| Rank 5 | pedinophyllols | 1.20 | flemingins | 1.01 |

Figure 4: Chemical structures for Aphanamixoids and Turrapubins, respectively

whereas $\beta$-furan is directly linked to cyclopentane derivatives in the other two cases. These results indicate that our system is capable of associating unknown compounds to their known analogues in our training dataset, consistent with human judgment.

By the same token, the average intra-cluster Tanimoto score of the cluster containing teuvissides (teuv) A, B, C, D, E, F, G and H is 99.3, and the cluster containing oleraceins (oleo) O, K and L is 99.6 (Lv et al., 2014; Jiao et al., 2015). The average intra-cluster Tanimoto score of the cluster containing sophodibenzosides (soph) A and B is 98.0, and the cluster containing flemingins (flem) A, B, C and O is 98.5 (Shen et al., 2013; Gumula et al., 2014). In contrast, the inter-cluster Tanimoto scores of the four clusters are $T_{(teuv.-oler.)} = 68$, $T_{(teu.-sophod.)} = 65$ or $T_{(teuv.-flem.)} = 57$. In this case, the Tanimoto similarity score matches the results of our system. Empirically, teuvissides and oleraceins are both glycosylated coumaroyltyramines, while sophodienzosides are dibenzoyl glycosides. Again, our system works nicely with respect to detecting glycosides.

Therefore, our system not only can cluster HSQC NMR spectra based entirely on the chemical structure similarity, but also outperforms the Tanimoto algorithm in structure similarity scoring.

## 5 CONCLUSIONS AND FUTURE WORK

We have shown that siamese neural networks perform significantly better than PLSI and LDA for the task of learning a similarity metric of NMR spectra to assist structure determination in Natural Product research. Our model maps directly from NMR spectra into a cluster space, where nearby points in the space have similar chemical structures. This architecture allows newly-discovered compounds to be mapped into the same space, resulting in a list of similarly-structured compounds. This list of similar compounds then provides strong clues to the structure of the novel compound, reducing the number of further experiments required to identify the structure.

In future work, we will apply hyperparameter optimization techniques to improve the architecture. In particular, we will apply the Bayesian hyperparameter tuning system called Hyperopt (https://github.com/hyperopt/hyperopt). We will also explore using an autoencoder on top of the cluster space, which is known to improve performance (Yann LeCun, personal communication). Finally, we have developed a soft purity loss that can be optimized by gradient descent, which we will use in combination with the contrastive loss as a regularizer.

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
