# OpenReview forum: "Using Deep Siamese Neural Networks to Speed up Natural Products Research"
_ICLR.cc/2019/Conference_

### Official Review · AnonReviewer1 · 2018-11-02
**Paper is more suitable for conferences/journals in chemistry or bioengineering**

**Rating:** 4
**Confidence:** 4

**Review:**

The paper proposes a Siamese network model for mapping the space of natural compounds to a latent representation space; furthermore, it utilizes this representation to compute a similarity score between an already known compound and a newly discovered one.

Some comments:
- Comparison with LDA requires more details: have you used the same number of topics for both small and large datasets?  How are you training it?
- Page 4 the definition of micro-averaging is missing: “Micro-averaging for precision and recall is formally defined as: “
- Figure 3 does not show a well defined clusters of various compound families because it’s using same color for different families in some cases (according to the caption of Figure 3). I wonder if you can somehow show for which compound families the colors have been repeated or maybe show fewer compound families.

The problem of finding similar compounds to a novel compound from NMR spectra is an interesting applied problem; however, technical novelty of the paper is not significant. Given the level of technical novelty, I believe the paper is more suitable for a more applied conference/journal in the fields of chemistry or bioengineering.

---

### Official Review · AnonReviewer3 · 2018-11-02
**Results are interesting but lack of novelty in the methodology**

**Rating:** 3
**Confidence:** 2

**Review:**

I didn’t worked in the field of structure elucidation from NMR spectroscopy so I might missed something. This paper utilized Siamese neural networks (Bromley 1993) and contrastive loss (Hadsell et al 2006) to learn a latent representation from the NMR spectra, which can be used for similarity search and the most similar compounds will be able to shad some light for the structures of the unknown natural product. The experiments showed significant improvement (AUC under precision-recall curve) over competitors.

My main concern is the lack of novelty in the methodology. The formulation is the same as Hadsell et al 2006 and the only change is an coefficient added to a loss term which kept to be fixed without explanation. As a result, I don’t feel this is enough to make it publishable at ICLR.

Some detailed comments below:
1. In eq(3), why P = 1.5? Some intuition or explanation?
2. I spent some time understanding the evaluation method in Section 4.2 and still not very sure I understand it. What is the formulae to compute your Tanimoto scores? Is this the same as PubChem Tanimoto scores which depends on fingerprints?
3. There is a formatting issue for the prediction and recall formulae.
4. I know in the computational mass spectrometry community there was an recent paper, “Critical Assessment of Small Molecule Identification 2016: automated methods“ where the winners were predicting the molecular fingerprints which directly shad the lights on the compound structures and can also be used for similarity searching, is this an option here for NMR data?

---

### Official Review · AnonReviewer2 · 2018-11-20
**Interesting topic - comparative analysis of different network architectures could improve the manuscript.**

**Rating:** 4
**Confidence:** 4

**Review:**

This paper identifies structure determination of novel natural products as a bottleneck in the drug discovery pipeline. The authors address this problem by using deep siamese neural networks to learn an representation of 2D NMR spectra that facilitates the rapid comparison of spectra for novel compounds with a database containing the spectra of molecules of known structure.

The authors have laboriously collected spectra from the published literature, using a protocol that included manual processing and orientation steps, resulting in a set of 1,385 NMR spectra from 104 families after imposing a minimum requirement of 10 compounds per family. They then implement two baseline topic modeling techniques - PLSI and LDA, and compare the performance of these baselines to the deep siamese CNN that they develop.

Using the random test/val/train split, the deep siamese CNN does show improved performance over the baseline models. However, the interesting and relevant use case is that in which the data is not randomly split, but rather complete families are held out from the training set for use in testing. The authors address this task by building a split in which they hold out four families: the aphanamixoids, teuvissides, tasiamides and macrolactins.

To evaluate performance, they calculate the averaged Tanimoto score for the top five closest compound families of these probe families. Unfortunately, the authors were not able to carry out this calcuation for two of the four probe families, restricting the evaluation to the aphanamixoids and teuvissides (I am confused as to why they couldn't calculate these scores - they appear to be relying on a PubChem server?). They compare these Tanimoto scores to the similarity score of the NMR spectra generated by their NN model. The similarity score either matches (in one case) or differs (in the other case) from the Tanimoto score, and the authors make the argument that this performance is correct.

I am confused that there is no requirement or criteria for the NN to detect the presence of a new compound family, which seems to be what this tasks calls for. This evaluation is rather limited, given that TC scores could not be computed for half of the probe families. There is also no comparison to the baseline models for this task.

Overall I think that this manuscript does a good job of identifying an interesting question, and makes a start at answering the question. To improve the manuscript I would ask that the authors carry out a more comprehensive evaluation of the performance using splits in which whole families are held out from the training set. Different splits might be evaluated, and in addition the model should be required to 'call' when it believes that a new compound family is present. Furthermore, it would be interesting to compare the siamese CNN approach to other network architectures. In addition, the performance of baseline models on the hold-family-out split should be assessed.

---

### Meta-Review · Area_Chair1 · 2018-12-14
**Better suited for another venue**

**Confidence:** 5
**Recommendation:** Reject

**Metareview:**

The reviewers highlighted that the application in the paper is interesting, but note a lack of new methodology, and also highlight serious flaws in the testing methodology. Specifically, the reviewers are discouraged by the straightforward reuse of Siamese networks without clear modifications. Further, the testing setup might be unfairly easy, since chemical families are represented in both training and test sets, while in true application of the method would be exposed to previously unseen chemical families.

The authors did not participate in the discussion, and address concerns. The reviewer consensus is a rejection.